# Consensus Statement on the Use of Near-Infrared Fluorescence Imaging during Pancreatic Cancer Surgery Based on a Delphi Study: Surgeons’ Perspectives on Current Use and Future Recommendations

**DOI:** 10.3390/cancers15030652

**Published:** 2023-01-20

**Authors:** Lysanne D. A. N. de Muynck, Kevin P. White, Adnan Alseidi, Elisa Bannone, Luigi Boni, Michael Bouvet, Massimo Falconi, Hans F. Fuchs, Michael Ghadimi, Ines Gockel, Thilo Hackert, Takeaki Ishizawa, Chang Moo Kang, Norihiro Kokudo, Felix Nickel, Stefano Partelli, Elena Rangelova, Rutger Jan Swijnenburg, Fernando Dip, Raul J. Rosenthal, Alexander L. Vahrmeijer, J. Sven D. Mieog

**Affiliations:** 1Department of Surgery, Leiden University Medical Center, 2333 ZA Leiden, The Netherlands; 2ScienceRight Research Consulting, London, ON N6A 3S9, Canada; 3Department of Surgery, University of California, San Francisco, CA 94143, USA; 4Department of General and Pancreatic Surgery, The Pancreas Institute, 37134 Verona, Italy; 5Research Institute against Cancer of the Digestive System (IRCAD), 67091 Strasbourg, France; 6Department of Surgery, Fondazione IRCCS Cà Granda Ospedale Maggiore Policlinico di Milano, 20122 Milano, Italy; 7Department of Surgery, University of California, San Diego, CA 92093, USA; 8Pancreas Translational & Clinical Research Center, Vita-Salute San Raffaele University, 20132 Milan, Italy; 9Department of Surgery, University of Cologne, 50923 Köln, Germany; 10Department of General, Visceral and Pediatric Surgery, University of Göttingen, 37075 Goettingen, Germany; 11Department of Visceral, Transplant, Thoracic and Vascular Surgery, University Hospital of Leipzig, 04103 Leipzig, Germany; 12Department of General, Visceral and Transplantation Surgery, Heidelberg University Hospital, 69120 Heidelberg, Germany; 13Hepato-Biliary-Pancreatic Surgery Division, Department of Surgery, Graduate School of Medicine, The University of Tokyo, Tokyo 113-8654, Japan; 14Department of Hepatobiliary and Pancreatic Surgery, Yonsei University College of Medicine, Seoul 03722, Republic of Korea; 15National Center for Global Health and Medicine, Tokyo 162-8655, Japan; 16Department of Upper Abdominal Surgery, Sahlgrenska University Hospital, 413 45 Gothenburg, Sweden; 17Department of Surgery, The Institute of Clinical Sciences, Sahlgrenska Academy, University of Gothenburg, 405 30 Gothenburg, Sweden; 18Department of Surgery, Amsterdam University Medical Center—Location AMC, 1105 AZ Amsterdam, The Netherlands; 19Cleveland Clinic Florida, Weston, FL 33331, USA

**Keywords:** fluorescence-guided surgery, intraoperative imaging, pancreatic cancer, cancer surgery, near-infrared fluorescence, indocyanine green, consensus, Delphi

## Abstract

**Simple Summary:**

Despite the potential of fluorescence imaging during pancreatic cancer surgery, more research is needed to facilitate the approval of tumor-targeted probes, standardize imaging techniques, and most importantly, gain trust from surgeons. Despite advancements in the development of novel probes, preclinical research settings do not always accurately represent the surgical setting. This first-of-its-kind Delphi consensus survey highlights current experiences and attitudes towards fluorescence imaging during pancreatic cancer surgery, specifically from surgeon’s perspectives. The results from this consensus survey highlight potential new directions for future research, which could facilitate the standardized use of fluorescence imaging during pancreatic surgery.

**Abstract:**

Indocyanine green (ICG) is one of the only clinically approved near-infrared (NIR) fluorophores used during fluorescence-guided surgery (FGS), but it lacks tumor specificity for pancreatic ductal adenocarcinoma (PDAC). Several tumor-targeted fluorescent probes have been evaluated in PDAC patients, yet no uniformity or consensus exists among the surgical community on the current and future needs of FGS during PDAC surgery. In this first-published consensus report on FGS for PDAC, expert opinions were gathered on current use and future recommendations from surgeons’ perspectives. A Delphi survey was conducted among international FGS experts via Google Forms. Experts were asked to anonymously vote on 76 statements, with ≥70% agreement considered consensus and ≥80% participation/statement considered vote robustness. Consensus was reached for 61/76 statements. All statements were considered robust. All experts agreed that FGS is safe with few drawbacks during PDAC surgery, but that it should not yet be implemented routinely for tumor identification due to a lack of PDAC-specific NIR tracers and insufficient evidence proving FGS’s benefit over standard methods. However, aside from tumor imaging, surgeons suggest they would benefit from visualizing vasculature and surrounding anatomy with ICG during PDAC surgery. Future research could also benefit from identifying neuroendocrine tumors. More research focusing on standardization and combining tumor identification and vital-structure imaging would greatly improve FGS’s use during PDAC surgery.

## 1. Introduction

Pancreatic cancer is a highly fatal malignancy with an average 5-year survival rate of below 5% [1]. This is largely due to its typically asymptomatic onset, resulting in most patients being diagnosed in late stages. Surgical resection combined with systemic treatment offers the only chance of possible cure. However, nearly 80% of patients present with unresectable disease at diagnosis due to extensive vascular tumor infiltration or distant metastases. During exploratory laparotomy, up to 38% of these patients already have occult metastases or an unresectable primary tumor [2]. Staging laparoscopies have been advised prior to laparotomies to spare patients from major surgery; but, despite careful patient selection for surgical resection predominantly by computed tomography (CT), metastases still are identified intra-operatively. Moreover, resection with tumor-positive margins (R1) still occurs in up to 50% of patients [3], leading to postoperative emergence of distant metastases and high recurrence rates [4,5]. Neoadjuvant therapy is increasingly being implemented to improve oncological outcomes and increase complete resection rates; however, CT scans struggle to evaluate vascular tumor involvement and to differentiate between viable cancer and therapy-induced fibrosis and necrosis [6,7]. This not only limits the prediction of resectability but poses challenges for surgeons while operating [8]. The ability to distinguish between tumor-positive and tumor-negative tissue is critical for successful oncologic surgery and microscopically radical resection (R0). Fortunately, in recent decades, novel techniques and advanced equipment have emerged that may improve the visualization of anatomical structures and completeness of resections.

One such technique is intraoperative near-infrared fluorescence (NIRF) imaging, also known as fluorescence-guided surgery (FGS). FGS enhances the identification of cancerous lesions, metastatic spread and major vascular structures, the evaluation of blood perfusion, the dissection of appropriate lymph node basins, and the attainment of tumor-free margins during solid cancer surgery [9]. FGS uses contrast agents with fluorescent characteristics in the NIR region (λ = 700–900 nm), which are visualized by NIR camera systems and displayed on a monitor in real time to the surgeon. FGS can be either targeted or non-targeted. Targeted FGS uses tumor-targeted NIRF probes containing targeting moieties—such as antibodies, peptides, or ligands—which bind with high affinity to proteins or receptors overexpressed on tumor cells and absent on adjacent normal cells. These targeting moieties are conjugated to a fluorophore that emits light in the NIR region [10]. An ideal NIRF probe would bind with high selectivity to all pancreatic cancer cells, including affected lymph nodes and distant metastases, but not to normal, inflamed, necrotic, or fibrotic tissue cells [11,12,13].

Non-targeted FGS mostly employs indocyanine green (ICG), which is currently the most commonly used fluorophore. Its absorption peak lies between 790 and 810 nm and emits fluorescence between 820 and 830 nm. Since its approval for clinical use in 1956, ICG has been applied in various branches of surgery and is now routinely used for angiographies, determining cardiac output, assessing hepatic function, and evaluating liver and gastric blood flow. Since the emergence of FGS, the applications of ICG have expanded to mapping sentinel lymph nodes and/or tumors during surgery for breast, lung, liver, and colon cancer, to identifying anastomotic leaks, and to assessing visceral perfusion. Its use has also expanded to endoscopic liver and biliary surgery and has shown potential during staging laparoscopies to detect radiologically occult metastases [2,14]. Despite the increasing use of ICG for a variety of applications, there remains a lack of consistency in terms of dose and administration time [15]. Additionally, its effectiveness during pancreatic cancer surgery is limited due to its non-targeted nature [16]. Although studies have investigated using ICG for targeted FGS for pancreatic cancer tumors, due to its enhanced-permeability and retention (EPR) effect, Hutteman et al. observed no useful tumor localization in seven out of eight patients, with equivalent ICG uptake in the tumor and healthy pancreas, and no EPR effect observed [17]. 

Despite significant efforts, no tumor-targeted NIRF probes have yet been approved for clinical use during pancreatic cancer surgery. Although it is important to investigate the (bio)chemical, pharmacokinetic, and pharmacodynamic aspects of novel fluorescent probes or dyes before clinical translation, it is critical to remember that preclinical research settings do not always accurately represent the surgical setting. Ultimately, the surgeon must benefit from these advancements to enhance patient outcomes. For this reason, a Delphi study was conducted to acquire a better understanding of experience with and attitudes toward FGS for pancreatic cancer, both with and without ICG, and how these might lead to future research recommendations. Our study is unique in that it focuses solely on surgeons’ perspectives, thereby guiding future research efforts targeting clinical effectiveness.

## 2. Materials and Methods

### 2.1. Delphi Survey

The Delphi technique is a well-established, multi-staged survey used to methodically gather judgments from an expert panel under the assumption that group opinion is more valid than individual opinion. A Delphi aims to achieve consensus and identify areas of non-consensus from relative experts on a particular topic through at least two rounds of anonymous surveys [18]. In these surveys, experts are asked for their opinion on statements in the first round and subsequently indicate their agreement or disagreement on cumulative data in subsequent rounds. For this Delphi, a three-round survey was conducted electronically via Google Forms survey software, in accordance with guidelines published by Keeney et al. [18]. All responses in each round remained anonymous. Individual responses were only known by the study moderator. 

### 2.2. Expert Panel

The expert panel invited to participate in the survey consisted of 38 surgeons selected to suit the purpose of the study and both contribute to and benefit from the discussion. All experts were identified by word of mouth from members of the International Society for Fluorescence Guided Surgery (ISFGS) and other experts. To be considered an expert, all invitees had to be pancreatic cancer surgeons, have extensive experience, knowledge, publication history, and a notable reputation either performing FGS or in FGS research, and be fluent in English. Due to the limited role of ICG in pancreatic cancer, experience with ICG was not a prerequisite to be considered an expert. All experts were identified by SM, AV, LM, KW, and TH, whom are all leading experts in the field themselves and are familiar with the reputations of the invited experts from previous FGS-related meetings, collaborations, research projects, publications, and clinical studies. 

### 2.3. Survey Development

The survey started with nine general demographic, surgical practice, and experience questions, followed by 20 statements, previously developed by Dip et al. [15], referring to the need for specific consent, contraindications to ICG use, and training in ICG use. These statements were included to gather baseline knowledge. For the remaining 47 statements and questions, PubMed was searched for publications that addressed specific clinical questions about FGS for pancreatic cancer. These publications were used to gain a better understanding of the most important topics for the Delphi to address and generate appropriate statements. Round 1 also included three open-ended and two multiple-choice questions to allow experts to raise any relevant issues not addressed in previous statements. The responses, once analyzed, served as a steppingstone for the subsequent rounds.

### 2.4. Survey Rounds and Statements

A personalized invitation was sent via email to 38 experts worldwide to request their participation in the Delphi survey, including a link to the first-round survey. Round 1 consisted of 76 items: nine questions concerning demographics, surgical practice, and experience, three open- and two multiple-choice questions, and 62 consensus statements on the general use of ICG and tumor-targeting probes during FGS for pancreatic cancer surgery. The 62 consensus statements and five above-noted questions were further divided into five modules: patient preparations and contraindications (n = 11), logistics of performing FGS (n = 13), benefits and drawbacks of FGS (n = 24), where to incorporate fluorescence imaging during surgery (n = 11), and future research (n = 8). The five open-ended and multiple-choice questions allowed for the collection of opinions on certain topics, resulting in the creation of 14 new consensus statements that were added in Round 2. Including these 14 new statements, a total of 76 consensus statements were voted on. Experts were asked to vote again in the following round for statements on which no consensus was reached, following published guidelines [18]. A 3rd round was conducted for new statements added in Round 2 where no consensus was reached. The cumulative results of previous rounds were displayed beneath each statement for all voters to see.

### 2.5. Consensus Process

Following published guidelines [18], an a priori decision was made to consider 70% inter-voter agreement evidence of consensus, and 80% participation in voting on each individual statement was considered evidence of a robust vote. To reduce the risk of agreement bias, some statements were worded favorably and others unfavorably towards FGS and/or the use of ICG.

## 3. Results

### 3.1. Expert Panel

Ultimately, 18 of the 38 invited experts participated in the Delphi survey (47%). The final panel included four surgeons from Germany, four from Italy, three from Japan, three from The Netherlands, two from the USA, and one each from South Korea and Sweden. Although two experts had not performed FGS, they were pancreatic cancer surgeons with sufficient knowledge on the topic, and their extensive experience and sizable publications relating to FGS research verified their eligibility. In addition, all final panel members were acknowledged by other experts as being international experts in their field. All participants were employed at an academic hospital specialized in pancreatic cancer surgery; however, FGS was only performed in hospitals of 13 participants. Further characteristics are summarized in Table 1.

### 3.2. Delphi Results

The results for all statements and questions in each round were considered robust (≥80% of eligible participants voting), as the lowest number of votes was 16/18. Consensus of ≥70% was reached for 61 of 76 statements (80.3%): 33 of 62 (53.2%) in the first round, 27 of 43 (62.8%) in the second round, and 1 of 2 (50%) in the third. Statements for which consensus was reached included: 7 of 11 (63.6%) statements in Module I (Table 2) on patient preparation and ICG contraindications, 12 of 13 (92.3%) statements in Module II (Table 3) regarding logistics performing FGS for pancreatic cancer, 23 of 30 (76.7%) statements in Module III (Table 4) regarding benefits and drawbacks of FGS for pancreatic cancer, 11 of 12 (91.6%) statements in Module IV (Table 5) regarding where fluorescence imaging is needed during pancreatic cancer surgery, and 8 of 10 in Module V (Table 6) on future research. Among the statements where consensus was reached, consensus ranged from 70.6–100%. 

### 3.3. General Statements Regarding ICG

All experts unanimously agreed that the general use of ICG is safe with very few side effects, though 83.3% felt patients should be asked about possible allergies before ICG administration. Regarding informed consent, 77.8% of experts agreed that the inability to attain informed consent should not be an absolute contraindication to FGS with ICG; however, no consensus was reached regarding the need for informed consent prior to FGS, with or without ICG. With 88.9–100% consensus, all experts agreed that the dose, concentration, and timing of ICG administration are very important. The consensus optimal dose for tumor imaging was ≤5 mg, which should be determined on a mg/kg basis, and the agreed-upon optimum timing of administration was >1 min before surgery.

### 3.4. ICG Use during Pancreatic Cancer Surgery

The experts unanimously agreed that ICG not being selective for pancreatic cancer is a limitation of FGS, and that research is necessary to determine the optimum dose, concentration, and timing for ICG use during pancreatic cancer surgery. Nonetheless, 88.9% felt a second intravenous dose of ICG could be given intra-operatively to better visualize pancreatic tumors. Aside from (primary) tumor imaging, 83.3% agreed that administering ICG 24 h before surgery might identify hepatic micro-metastases. Almost all (94.12%) agreed that ICG can evaluate blood flow during organ-preserving surgical techniques—such as the Warshaw technique, spleen-preserving distal pancreatectomy (SPDP), and duodenum-preserving pancreatic head resection (DPPHR)—and is advantageous during pancreatic cancer surgery.

### 3.5. Fluorescence Imaging during Pancreatic Cancer Surgery

Most (88.9%) experts disagreed with the statement that “intraoperative frozen section analysis is insufficient for identifying resection margins, but fluorescence imaging is”, and 77.8% agreed that though intraoperative frozen section analysis is sufficient, precision analysis can be enhanced by integrating FI into the workflow. Regarding benefits and drawbacks, most experts agreed, with >80% consensus, that fluorescence imaging is useful when visual inspection and palpation are limited, that there are no disadvantages to its use during pancreatic cancer surgery, that it improves intra-operative visualization and is of added benefit during pancreatic cancer surgery, and that real-time flow assessments help to avoid confirmation bias. Between 70–80% agreed that FI (including its equipment) does not interfere with surgical workflow, is easy to use, and neither increases nor decreases the rate of complications, and 72.2% disagreed that FI is unable to distinguish between viable tumor tissue and neoadjuvant therapy-induced necrosis/fibrosis. On the other hand, 94.4% agreed that FI has limited penetration depth, 77.8% that inadequate empirical evidence supporting its efficacy is a major barrier to adopting FI during pancreatic cancer surgery, 72.2% that one limitation is the false positive/false negative fluorescence that may result depending on the distance between the tip of the camera and target tissue, and that FGS still being experimental is a limitation. Lastly, 83.3% agreed that another limitation of FGS for pancreatic cancer is that different pancreatic tumors, such as pancreatic ductal adenocarcinoma (PDAC) versus pancreatic neuroendocrine tumors (panNET), may have different fluorescent features.

### 3.6. Where Fluorescence Is Needed and Future Recommendations

Module IV (Table 5) concerns where our experts felt fluorescent guidance is needed most during pancreatic cancer surgery, with either ICG or tumor-targeted probes. Over 80% agreed that fluorescence is needed to visualize the anatomy of the extra-hepatic bile duct during SMA lateral border dissection and to detect and accurately localize metastatic lesions, to determine accurate resection margins, and to visualize surrounding area structures such as the biliary ducts and lymph nodes. Between 75–80% consensus was reached on incorporating fluorescence for accurately localizing lesions, determining extra-pancreatic spread, visualizing vascular structures such as the SMA or SMV, distinguishing between viable tumor tissue and neoadjuvant therapy-induced necrosis/fibrosis, and determining the viability of anastomoses and surrounding organs (e.g., colon, stomach, spleen).

## 4. Discussion

This Delphi demonstrates a high degree of consensus among experts on the safety and potentials of FGS for pancreatic cancer with ICG and tumor-targeted probes. However, despite perceived benefits, experts agreed that FGS, specifically with tumor-targeted probes, should not yet be implemented into routine use due to insufficient empirical evidence proving its benefit over standard back-table methods, and the absence of clinically available tumor-targeted NIRF probes. This study also identified novel directions for future research, such as developing tumor-targeted probes for primary tumor identification in combination with ICG to visualize vasculature and anatomical structures, including pNETs in future research, and standardizing FGS protocols. Achieving these objectives would likely increase surgeons’ willingness to integrate fluorescence imaging into their standard workflow.

ICG has been investigated extensively for FGS, and its safety and effectiveness have been demonstrated in multiple publications [14,19,20,21,22]. This is mirrored in our survey’s findings, which included consensus that ICG is safe and should not be presented as experimental. Although dose, concentration, and timing of ICG administration are critical, all experts agreed that further research is needed to achieve uniformity and determine optimal thresholds. These results are consistent with those of another recently published Delphi survey on the general use of fluorescence imaging and ICG [15]. 

Notwithstanding the insights gained from this study, our Delphi’s fundamental limitations are its subjective nature and limited sample size. However, though the results of a Delphi do not reveal “correct” responses, its conclusions do point to truths stemming from the experience of international experts. Although inviting only relevant experts to participate may introduce bias in favor of any subject, the skepticism that can be seen in our results demonstrates experts’ ability to remain impartial. Furthermore, though the size of a Delphi panel can be as small as three members [18], a larger expert panel could have improved both the accuracy and generalizability of consensus. Still, the experts in this Delphi were geographically diverse, well published, and highly experienced. Similarly, though two experts have not performed any surgeries with fluorescence, their credibility and experience in this field supports their eligibility.

Although ICG has been observed to accumulate in some solid tumors through the EPR effect [23], pancreatic tumors exhibit minimal to no EPR effect due to their highly stromal nature [24], and studies investigating the role of ICG for pancreatic cancer imaging have failed to document any useful tumor demarcation [17]. Similarly, our experts acknowledged that ICG’s lack of specificity for pancreatic cancer is a clear limitation of FGS. These considerations suggest diverting from the use of ICG to visualize pancreatic tumors and further pursue the development and translation of tumor-targeted probes [25,26].

However, this shift does not apply to all aspects of FGS for pancreatic cancer, as our panel viewed ICG as useful for more than just identifying lesions. It was agreed, with strong consensus, that ICG also aids in evaluating blood flow during organ-preserving surgical techniques such as the Warshaw, SPDP, and DPPHR, and that administering ICG at least 24 h before surgery can assist in identifying liver micro-metastases. The latter is especially important, as liver metastases in pancreatic cancer patients are an important prognostic factor, and preoperative imaging modalities—such as CT and magnetic resonance imaging (MRI)—are relatively insensitive to detecting micro-metastases [27]. For this purpose, ICG can be administered prior to a staging laparoscopy to identify otherwise-undetected liver metastases [2,28]. In addition, visualizing the extra-hepatic bile duct, distinguishing between viable tumor tissue and neoadjuvant therapy-induced necrosis/fibrosis, visualizing surrounding structures and vasculature, determining the viability of anastomoses and surrounding organs, and enhancing visualization during SMA lateral border dissection all were surgical steps that could appreciably benefit from fluorescence, according to our experts. The need for improved visualization during these steps has been mentioned repeatedly in various publications [8,29,30]. Consequently, though FGS with tumor-targeted probes could aid in detecting and resecting viable lesions, ICG could be used simultaneously to visualize surrounding structures and vasculature, confirm perfusion in anastomoses, and detect biliary leaks after reconstruction. This could improve resections and organ preservation and lower morbidity rates. 

This Delphi also identified new areas of research to explore. Most experts perceived the different fluorescent features of different pancreatic tumors—such as PDAC versus panNETs—as a limitation of FGS, attributable to the hypervascularity of panNETs. Considering that panNETs only account for 5% of tumors [31], pre-clinical research has primarily focused on PDAC. However, exact identification of a small panNET amendable to minimally invasive parenchyma-sparing resection (i.e., enucleation) can be challenging. FGS can be of benefit in such circumstances, as already shown with methylene blue and ICG [32,33,34]. Furthermore, the following statement, which was added based on the results of an open-ended question, achieved high consensus: “A limitation of FGS for pancreatic cancer is the false positive/false negative fluorescence that may result depending on the distance between the tip of the camera and target tissue”. Although the position of the camera is known to be important for obtaining the appropriate amount of signal [16,35], an abundance of fluorescence cameras are available from major brands, as demonstrated by the numerous camera systems our experts use, each having its own configurations and specifications. Although this is not the only cause of false positive/false negative results, standardized set-up, data acquisition, and reporting might contribute to reducing their occurrence. The various imaging systems used also result in considerable inter- and intra-institution reporting variance and can have critical effects during clinical trials or when seeking regulatory approval. As previously suggested, standardized methodologies would be of paramount importance during both processes [35], whether used during open, minimally invasive, or robotic surgery. 

Not only does FGS require a shift in research focus, but it also requires also increased trust from surgeons. Surprisingly, our experts could not agree on whether FGS should be incorporated into routine use for pancreatic cancer. The current standard for intraoperatively identifying resection margins is through frozen section analysis. However, frozen section analysis is time-consuming and sometimes lacks precision, so the introduction of real-time FI is important [36]. Despite considerable agreement on the benefits of FGS, surgeons still consider frozen section analysis sufficient for identifying resection margins and feel that integrating FI into the workflow merely *could* enhance precision. This could be because FGS is still seen as an emerging technology, which is apparent in Table 1, as 66.7% of experts have only been performing FGS for less than 5 years. This adds to the argument that, despite numerous efforts, investigators have not yet proven that FGS has any significant advantage over current methods. Fortunately, Module III (Table 4) highlights the perceived benefits and drawbacks of FGS, the majority of which have already been widely discussed [14,15,19,20,21,22], indicating growing belief in FGS.

## 5. Conclusions

Among international experts, FGS with both ICG and tumor-targeted probes is considered safe and has the potential to facilitate pancreatic cancer surgery, but standardized dosing, concentration, and timing remain lacking, and currently, there is no optimal tumor-targeted NIRF probe available for pancreatic cancer. Despite obvious benefits, FGS is currently perceived to only supplement established techniques such as frozen section analysis, and more research is needed before it can be considered a first-line approach. Our first-of-its-kind Delphi also suggests shifting current focus from single-modality applications to combining tumor-targeted probes and ICG, the combination of which could contribute to enhanced tumor resection and organ preservation and, most importantly, better outcomes.

## Figures and Tables

**Table 1 cancers-15-00652-t001:** Practice characteristics of experts.

Practice Characteristic	n	%
**Region of practice (n = 18)**		
Africa	0	0%
Asia-Pacific	4	22.2%
Europe	12	66.7%
Middle East	0	0%
North America	2	11.1%
South America	0	0%
**Country of practice (n = 18)**		
Germany	4	22%
Italy	4	22.2%
Japan	3	16.7%
The Netherlands	3	16.7%
South Korea	1	5.6%
Sweden	1	5.6%
USA	2	11.1%
**Nature of employment (n = 18)**		
Academic hospital	18	100%
Non-teaching academic hospital	0	0%
Non-teaching hospital	0	0%
**Hospital specialization for pancreatic cancer surgery (n = 18)**		
Yes	18	100.0%
No	0	0.0%
**Years performing pancreatic cancer surgery (n = 18)**		
Less than 10 years	3	16.7%
10–20 years	9	50.0%
More than 20 years	6	33.3%
**Years using fluorescence during pancreatic cancer surgery (n = 18)**		
Less than 5 years	12	66.7%
5–10 years	3	16.7%
More than 10 years	3	16.7%
**Availability/performance of FGS at hospital of employment (n = 18)**		
Yes	13	72.2%
No	5	27.8%
**Surgeries performed using fluorescence (n = 18)**		
0	2	11.1%
1–99	7	38.9%
100–200	5	27.8%
More than 200	3	16.7%
Unsure	1	5.6%
**Camera systems used (n = 18) ***		
Stryker	9	50.0%
Olympus	7	38.9%
Storz	4	22.2%
Quest Spectrum	4	22.2%
Da Vinci Firefly	3	16.7%
None	2	11.1%
Visionsense	2	11.1%
Arthrex	1	5.6%
HyperEye Medical System (HEMS)	1	5.6%
Medtronic	1	5.6%
Mini-FLARE	1	5.6%

FGS = fluorescence guided surgery; * = some experts used multiple imaging systems.

**Table 2 cancers-15-00652-t002:** Module I—Statements regarding patient preparation and ICG contraindications.

Statement	# Votes	Response	# Rounds	% Consensus
**CONSENSUS REACHED**				
**Using ICG**				
Allergic reactions to ICG are extremely rare.	18	Agree	1	100%
All patients should be asked about possible allergies to iodine, shellfish, or ICG prior to having ICG administered.	18	Agree	1	83.3%
Inability to provide informed written consent is an absolute contraindication to using ICG.	18	Disagree	2	77.8%
Prior to undergoing FGS with ICG, patients should be informed that its use is still experimental.	16	Disagree	1	75%
Suspected allergy to iodine or shellfish is a relative contraindication to FGS with ICG.	17	Agree	2	72.2%
Pregnancy is an absolute contraindication to FGS with ICG.	17	Agree	1	70.6%
**Using ICG and/or tumor-targeted probes**				
Inability to provide informed written consent is an absolute contraindication to FGS.	18	Disagree	2	77.8%
**NO CONSENSUS REACHED**				
**Using ICG**				
Prior to undergoing FGS, patients must provide informed written consent specific to its use.	17	Agree	2	64.4%
Prior to receiving ICG, patients must provide informed written consent specific to its use.	17	Disagree	2	61.1%
Prior to receiving ICG, patients should be provided with written information specifically addressing its use.	18	Disagree	2	58.8%
**Using ICG and/or tumor-targeted probes**				
Prior to undergoing FGS, patients should be provided with written information specifically addressing its use.	18	Disagree	2	61.1%

ICG = indocyanine green; FGS = fluorescence-guided surgery.

**Table 3 cancers-15-00652-t003:** Module II—Statements regarding logistics of performing FGS for pancreatic cancer.

Statement	# Votes	Response	# Rounds	% Consensus
**CONSENSUS REACHED**				
**Using ICG**				
For FGS with ICG, the timing of ICG administration (how long before surgery) is very important.	18	Agree	1	100%
Research is necessary to determine the optimum dose and concentration of ICG and timing of ICG administration for pancreatic cancer surgery.	18	Agree	1	100%
For FGS with ICG, the dose of ICG administered is very important	18	Agree	1	88.9%
A second intravenous dose of ICG can be given intra-operatively to better visualize pancreatic tumors.	17	Agree	2	88.9%
When using ICG during pancreatic cancer surgery, the optimum dose to administer is...	16	5 mg or less	2	83.3%
For FGS with ICG, the concentration of ICG administered is very important.	18	Agree	1	83.3%
After administration, ICG becomes visible in the pancreas within seconds.	17	Agree	1	82.4%
The dose of ICG to administer for FGS should be determined on a mg-per-kg basis or as an absolute dose.	18	mg/kg	2	77.8%
When using ICG, the optimum timing for ICG administration prior to pancreatic cancer surgery is…	17	>1 min before	1	70.6%
**Using ICG and/or tumor-targeted probes**				
Intraoperative frozen section analysis is NOT sufficient for identifying resection margins, but fluorescence imaging is.	18	Disagree	1	88.9%
Intraoperative frozen section analysis is sufficient for identifying resection margins; but the precision of this analysis CAN be enhanced by integrating fluorescence into the workflow.	18	Agree	1	77.8%
Neither intraoperative frozen section analysis nor fluorescence imaging sufficiently identifies resection margins.	18	Disagree	2	77.8%
**NO CONSENSUS REACHED**				
**Using ICG and/or tumor-targeted probes**				
Intraoperative frozen section analysis is sufficient for identifying resection margins, whereby fluorescence imaging yields NO additional benefit.	17	Disagree	2	66.7%

FGS = fluorescence-guided surgery; ICG = indocyanine green; mg = milligram; kg = kilogram; min = minute.

**Table 4 cancers-15-00652-t004:** Module III—Statements regarding benefits and drawbacks of FGS for pancreatic cancer.

Statement	# Votes	Response	# Rounds	% Consensus
**CONSENSUS REACHED**				
**Using ICG**				
* A limitation of FGS for pancreatic cancer is that ICG is not selective for pancreatic cancer tissue.	17	Agree	1	100%
When using ICG, the background fluorescence of tissue surrounding large arteries and veins is bothersome.	17	Agree	2	94.4%
* ICG can evaluate blood flow during organ-preserving surgical techniques—such as the Warshaw, SPDP, and DPPHR—and is advantageous during pancreatic cancer surgery.	17	Agree	1	94.1%
* If ICG is applied 24 h before surgery, micro-metastases in the liver might become evident.	18	Agree	1	83.3%
**Using ICG and/or tumor-targeted probes**				
FGS has a limited penetration depth.	18	Agree	1	94.4%
FGS is useful when visual inspection and palpation are limited (e.g., minimally invasive surgery).	18	Agree	1	94.4%
*Real-time flow assessment helps avoid confirmation bias.	17	Agree	2	94.1%
There are no disadvantages of FGS for pancreatic cancer.	18	Agree	1	88.9%
Fluorescence imaging improves/worsens intraoperative visualization.	17	Improves	1	88.2%
There are no advantages of FGS for pancreatic cancer.	17	Disagree	1	88.2%
Fluorescence imaging is of added benefit during pancreatic cancer surgery.	18	Agree	1	83.3%
* A limitation of FGS for pancreatic cancer is that different pancreatic tumors (PDAC vs. panNETs) may have different fluorescent features.	18	Agree	1	83.3%
Fluorescence imaging improves/worsens decision making.	17	Improves	1	82.4%
FGS equipment has low image quality.	17	Disagree	1	82.4%
FGS allows for more radical resections.	16	Disagree	2	77.8%
Inadequate empirical evidence supporting its efficacy is a major barrier to implementing the use of FGS for pancreatic cancer.	18	Agree	1	77.8%
Background fluorescence of clearance routes is bothersome.	17	Agree	1	76.5%
* A limitation of FGS for pancreatic cancer is the false positive/false negative fluorescence that may result depending on the distance between the tip of the camera and target tissue.	18	Agree	1	72.2%
FGS is unable to distinguish between viable tumor tissue and neoadjuvant therapy-induced necrosis/fibrosis.	17	Disagree	2	72.2%
Fluorescence imaging (including equipment) does/does not interfere with surgical workflow.	18	Does not	1	72.2%
FGS results in a decreased/similar/increased rate of complications.	18	Similar	2	72.2%
* One limitation of FGS is that it is still experimental.	18	Agree	1	72.2%
FGS equipment is easy/difficult to use.	17	Easy	1	70.6%
**NO CONSENSUS REACHED**				
**Using ICG and/or tumor-targeted probes**				
FGS should be implemented as routine use for pancreatic cancer.	18	Agree	2	66.7%
The ambient room lighting required during procedures during FGS is bothersome.	17	Agree	2	61.1%
FGS results in over-reliance on the fluorescence signal.	17	Disagree	2	61.1%
FGS is unable to distinguish between tumor and surrounding stroma.	17	Disagree	2	61.1%
FGS has a minimal/average/steep learning curve	17	Minimal	2	55.6%
FGS has an increased operating room time.	18	Disagree	2	55.6%
Identifying suitable surgical candidates who might benefit from FGS for pancreatic cancer is a major barrier to its use during pancreatic cancer surgery.	17	Disagree	2	55.6%

FGS = fluorescence-guided surgery; ICG = indocyanine green; SPDP = spleen-preserving distal pancreatectomy; DPPHR = duodenum-preserving pancreatic head resection; FGS = fluorescence-guided surgery, PDAC = pancreatic ductal adenocarcinoma, panNETs = pancreatic neuroendocrine tumors; * = new statement added in round 2 based on open-ended questions in round 1.

**Table 5 cancers-15-00652-t005:** Module IV—Statements regarding incorporating fluorescence during pancreatic cancer surgery—where is it needed?

Statement	# Votes	Response	# Rounds	% Consensus
**CONSENSUS REACHED**				
**Using ICG and/or tumor-targeted probes**				
* To visualize the anatomy of the extra-hepatic bile duct.	18	Agree	1	94.4%
* SMA lateral border dissection (indirect pancreas enhancement).	18	Agree	1	83.3%
The detection and accurate localization of metastatic lesions.	18	Agree	1	83.3%
Visualization of surrounding structures (biliary ducts, lymph nodes).	18	Agree	1	83.3%
The determination of accurate resection margins.	18	Agree	2	83.3%
The accurate localization of lesions.	18	Agree	1	77.8%
The determination of extra-pancreatic spread.	18	Agree	1	77.8%
Visualization of vascular structures such as the superior mesenteric artery and vein.	18	Agree	2	77.8%
Distinguishing between viable tumor tissue and neoadjuvant therapy-induced necrosis/fibrosis.	18	Agree	2	77.8%
Determining the viability of anastomoses.	17	Agree	1	76.5%
Determining the viability of surrounding organs (colon, stomach, spleen).	17	Agree	1	76.6%
**NO CONSENSUS REACHED**				
**Using ICG and/or tumor-targeted probes**				
* Visualization of pancreatic juice leaking from the stump/anastomosis.	18	Agree	2	61.1%

SMA = superior mesenteric artery; * = new statement added in round 2 based on open-ended questions in round 1.

**Table 6 cancers-15-00652-t006:** Module V—Future research on FGS for pancreatic cancer.

Statement	# Votes	Response	# Rounds	% Consensus
**CONSENSUS REACHED**				
**Using ICG and/or tumor-targeted probes**				
More research should be dedicated to FGS to facilitate the eventual implementation of its routine use for pancreatic cancer.	18	Agree	1	100%
* Future research should focus on quantifying the degree of fluorescence intensity to give objective parameters indicating the viability of structures.	18	Agree	1	100%
* Future research should focus on combining hyperspectral or multispectral imaging with ICG-fluorescence imaging.	18	Agree	1	100%
* Future research should focus on increasing the specificity of fluorescent probes.	18	Agree	1	94.4%
In the future, fluorescence imaging should greatly simplify certain decision-making stages (i.e., a stage that requires considerable analysis and/or time, but whereby the mere sight/absence of a fluorescence signal is a clear indicator of how to proceed).	18	Agree	1	94.4%
Surgeons should be introduced to FGS during their residency training.	18	Agree	1	83.3%
* Future research should focus on decreasing/predicting the risk of pancreatic fistula.	18	Agree	1	77.8%
Not just surgical residents, but residents in other, non-surgical fields should be introduced to FGS during their residency training.	18	Agree	2	72.2%
**NO CONSENSUS REACHED**				
**Using ICG and/or tumor-targeted probes**				
Future research should shift away from identifying pancreatic tumors and focus more on other objectives, such as identifying other anatomical structures and assessing anastomoses.	18	Agree	2	66.7%
A physician trainee’s first introduction to FGS should begin during medical school/residency training	18	Residency training	2	55.6%

FGS = fluorescence-guided surgery; ICG = indocyanine green; mg = milligram; kg = kilogram; min = minute; * = new statement added in round 2 based on open-ended questions in round 1.

## Data Availability

The data presented in this study are available upon request from the first author. The data are not publicly available due to privacy reasons.

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
