# Peer review of "Consensus Statement on the Use of Near-Infrared Fluorescence Imaging during Pancreatic Cancer Surgery Based on a Delphi Study: Surgeons’ Perspectives on Current Use and Future Recommendations"

_cancers, 2023, doi:10.3390/cancers15030652_

Round 1
Reviewer 1 Report
This is a well written consensus statement paper regarding the utility of FGS using ICG in pancreatic surgery. Compared with the FGS in liver surgery, the usefulness has not been well established, but such immature status enhanced the importance of this paper clarifying the future issues to be addressed.
I have only a few comments as below.
1. It would facilitate the readers' understanding if the authors summarize the representative statements with high consensus rate with bullet points.
2. Regarding the ability of FI to distinguish between viable tumor tissue and neodadjuvant therapy-induced necrosis/fibrosis, please present severel citations showing this.
Author Response
Thank you for taking the time to review this manuscript, and for your feedback. Please see our responses to your comments below:
-
Although this was an original idea, considering the length of the manuscript we decided to opt out of bullet points to keep the structure concise. We feel the tables display all statements with sufficient clarity.
-
We do not state that FI is able to distinguish between NAT-induced necrosis/fibrosis, we only state that distinguishing between necrotic/fibrotic and healthy tissue is a recurring challenge with modern imaging technology and underline the fact that an ideal fluorescent probe is able to make his distinction. The experts also emphasize this need in Module IV: where fluorescence is needed and future recommendations (77.8% agreement). Although in module III, 72% disagree that “FGS is unable to distinguish between viable tumor tissue and neoadjuvant therapy-induced necrosis/fibrosis”, the experts are most likely referring to the use of tumor-targeted probes here as this statement falls under the heading “with ICG/tumor targeted probes”. This is their personal experience which cannot be shown through citations. In the discussion (line 324-330) we mention the latter statement and provide citations of other studies that have highlighted a need for improved visualization repeatedly.
Reviewer 2 Report
Overall, the authors present a very interesting topic that is certainly promising for the future. However, it is not clear to me in the manuscript what surgical volume defines an expert in pancreatic surgery. For example, there are surgeons other than the co-author at the University Hospital of Cologne who manage the Pancreas Center and thus should rather deserve the designation expert. Therefore, a sharper definition of the designation "expert" needs to be made in the material and methods section. The authors could use the number of annual pancreatic procedures and, especially in Germany, whether the experts are registered center surgeons of a certified pancreas center.
Author Response
Thank you very much for taking the time to review this manuscript, and for your feedback. Please see our response below:
We have added additional statements (line 147-148, line 198-199) to clarify our definition of experts. It is difficult to approach the definition in an institutional/regulatory way since we’re dealing with a global collection of experts whose countries and institutions all have different regulations and definitions. However, we feel confident that our panel is a strong representation of experts in the field for various reasons; they were identified by word of mouth from ISFGS members and other experts and by reviewing published articles on pancreatic cancer surgery and fluorescence, and all members of our panel have an academic appointment at a center specialized in pancreatic cancer surgery. When we look at published Delphi studies on fluorescence surgery, we mostly see the same experts in each respective list of authors who have all repeatedly been defined as appropriate experts. These experts also make up the majority of co-authors listed in this manuscript, and we regard these factors to be sufficient in defining our panel.
Reviewer 3 Report
Main Comments:
This manuscript deals with the use of near-infrared fluorescence imaging during pancreatic cancer surgery. It relies on the evaluation of subjective opinions and does not break new ground in terms of technical innovations. The presented results provide, however, an overview on the status quo and a basis for future investigation. In this context, this paper may be seen as a stimulus for further research.
Additional Comments/Suggestions:
Table 1: "Camera systems used (n=18)"; 4+2+1+1+1+4+1+1+1=16 – here the numbers are not clear for the reader.
Table 4: "A limitation of FGS for pancreatic cancer is that ICG is not as selective for pancreatic cancer tissue" – "not as selective" as what? (– or did you just mean: not selective…)
Discussion, line 336/337: "However, exact identification of a small panNET amendable for minimally invasive parenchyma-sparing resection (i.e., enucleation) can be challenging" -> However, exact identification of a small panNET amenable to minimally invasive parenchyma-sparing resection (i.e., enucleation) can be challenging.
Line 366: Please add a period/full stop after "FGS".
Reference list: The format of the journal names is not consistent.
Author Response
Thank you very much for taking the time to review this manuscript and for your feedback and attention to detail. Please see our responses below:
-
Thank you for pointing this out. Upon further inspection it became apparent these results were inaccurate. The results have been re-calculated and are now presented accurately in the table. The column titled “n” refers to the frequency, which for this question = 35. This is because the experts used multiple camera systems. An asterisk was added to the table to clarify this. We would also like to note that after discovering this error, all other tables were reanalyzed to check for any further miscalculations; none were found.
-
We meant “not as selective as tumor-targeted probes” but have changed this sentence to “not selective..” to maintain conciseness.
- This has been adjusted.
- This has been adjusted.
-
For this manuscript, Cancers’ own downloadable EndNote style file was used, but the software did not convert journal names to their correct abbreviations. All journal names have now been adjusted.
Reviewer 4 Report
Concerning the manuscript entitled "Consensus Statement on the Use of Near-Infrared Fluorescence Imaging During Pancreatic Cancer Surgery Based on a Delphi Study: Surgeons’ Perspectives on Current Use and Future Recommendations"
Surgery for PDAC still remains the primary choice but still has great need to be facilitated and optimized. While, as the authors state, "despite perceived benefits, experts agreed that FGS, specifically with tumor-targeted probes, should not yet be implemented into routine use due to insufficient empirical evidence proving its benefit over standard back-table methods, and the absence of clinically available tumor-targeted NIRF probes" this study provides important insights that will help future studies identify directions for research to better standardize, optimize and apply FGS and novel applications. Indeed, this study has identified new directions for future research.
The first of its kind manuscript is very well organized, presented and easy to read and understand. Further, I find no faults with the study methods and the auhtors are quite honest concerning its possible weeknesses and applicability.
This is an important consensus presentation that I believe warrants immediate publication
Author Response
Thank you very much and for taking the time to review this manuscript. Your feedback and appraisal is very much appreciated!
Round 2
Reviewer 2 Report
The authors responded to my questions satisfactorily